# SEE, THINK, ACT: ONLINE SHOPPER BEHAVIOR SIMULATION WITH VLM AGENTS

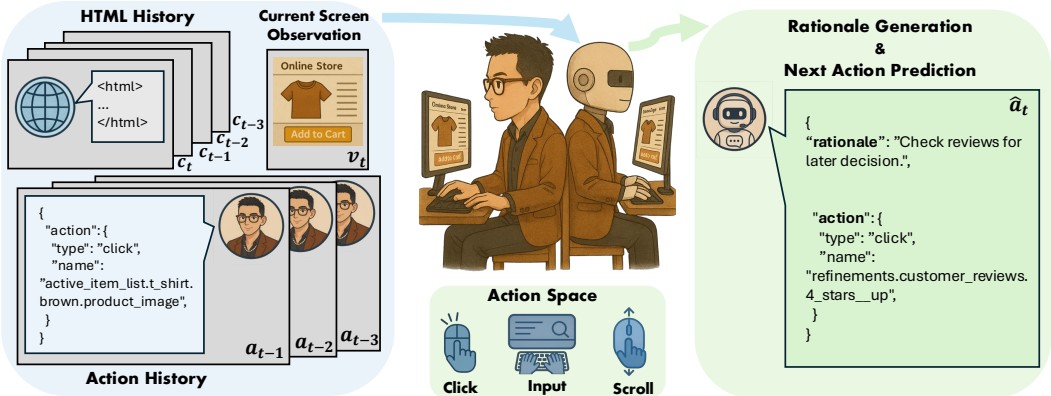

Figure 1: An overview of GUI-aware simulation of human web shopper behavior with a VLM agent. Given a sequence of past actions $a_{t-3...t-1}$ accompanied by corresponding website observations $c_{t-3...t}$, the model predicts the next action $\hat{a}_t$ and its underlying rationale $r_t$ by reasoning over the accumulated action history and the current website context, which includes both text-based HTML $c_t$ and image-based GUI screenshot $v_t$.

## ABSTRACT

Large Language Models (LLMs) have recently demonstrated strong potential in simulating online shopper behavior. Prior work has improved action prediction by applying supervised fine-tuning (SFT) on action traces with LLM-generated rationales, and by leveraging reinforcement learning (RL) to further enhance reasoning capabilities. Despite these advances, current approaches rely solely on text-based inputs (e.g., such as HTML content and action histories) and overlook the essential role of visual perception in shaping human decision-making during web GUI interactions. In this paper, we investigate the integration of visual information, specifically webpage screenshots, into behavior simulation via vision-language models (VLMs), leveraging the publicly available OPeRA dataset. By grounding agent decision-making in both textual and visual modalities, we aim to narrow the gap between synthetic agents and real-world users, thereby enabling more faithful and cognitively aligned simulations of online shopping behavior. Specifically, we employ SFT for joint action prediction and rationale generation, conditioning on the full interaction context, which comprises action history, past HTML observations, and the current webpage screenshot. To further enhance reasoning capabilities, we integrate RL with a hierarchical reward structure, scaled by a difficulty-aware factor that prioritizes challenging decision points. Empirically, our studies show that incorporating visual grounding yields substantial gains: the combination of text and image inputs improves exact match accuracy by more than 6% over text-only inputs. These results indicate that multi-modal grounding not only boosts predictive accuracy but also enhances simulation fidelity in visually complex environments, which captures nuances of human attention and decision-making that text-only agents often miss. Finally, we revisit the design space of behavior simulation frameworks, identify key methodological limitations, and propose future research directions toward building efficient and effective human behavior simulators. [1]

---

[1]The code and model checkpoints will be released upon paper acceptance.

# 1 INTRODUCTION

Simulating human behavior in web-based environments has emerged as a promising research direction, enabling a wide range of applications including digital assistant training, GUI design optimization, and large-scale user behavior forecasting (Yao et al., 2022; Achiam et al., 2023; Wang et al., 2025a; Zhang et al., 2024; Lu et al., 2025c; Chen et al., 2025; Jia et al., 2024; Koh et al., 2024a; Yu et al., 2024; Koh et al., 2024b; Gu et al., 2024; Ashraf et al., 2025). Recent advances in Large Language Models (LLMs) have demonstrated remarkable capabilities in this domain, offering fluent reasoning, contextual awareness. Researchers have begun leveraging LLMs to simulate human behavior in web-based environments, aiming to generate realistic human action sequences on digital platforms, which has promising applications across domains such as e-commerce (Lu et al., 2025a; Zhang et al., 2025; Kasuga & Yonetani, 2024; Khatuya et al., 2025; Wang et al., 2025c), education (Yao et al., 2021; Chu et al., 2022; Scarlatos et al., 2022), and social computing (Pan et al., 2006; Anthis et al., 2025; Mou et al., 2024). A growing body of work has focused on enhancing human behavior simulation performance in the web-based shopping scenario through LLM-based methods. One line of research augments training datasets with LLM-synthesized rationales to provide richer supervision signals and employs supervised fine-tuning (SFT) to improve action prediction accuracy (Lu et al., 2025a). Another complementary direction leverages reinforcement learning (RL) to align model-generated reasoning with realistic user trajectories, refining the model's ability to mimic decision-making patterns observed in human users (Zhang et al., 2025). However, these approaches share a fundamental limitation: they rely exclusively on text-based inputs such as HTML content and action histories. While textual signals are critical, they only provide a partial view of the online shopping experience. In contrast, real users heavily rely on visual perception when navigating and making decisions on modern, image-rich webpages (Agosto, 2002; Lurie & Mason, 2007; Chen et al., 2017; Zhang et al., 2023). Ignoring the visual modality hinders the model's ability to faithfully capture the full spectrum of user behavior, especially in tasks that require understanding product layouts, button salience, or the visual composition of search results (Linsley et al., 2017; 2018; Lin et al., 2025).

To bridge the gap between current text-only simulation methods and human decision-making processes, we incorporate visual information (e.g., webpage screenshots) into the behavior simulation pipeline. Specifically, we leverage vision-language models (VLMs) as a natural extension of large language models (LLMs) to jointly process textual and visual modalities (Alayrac et al., 2022; Bai et al., 2025). As illustrated in **Fig. 1**, the model input consists of a sequence of past actions $a_{t-3...t-1}$ together with the corresponding website observations $c_{t-3...t}$. Given this context, the model predicts the next action $\hat{a}_t$ and its associated rationale $r_t$ by reasoning over the accumulated action history and the current website state, which incorporates both the text-based HTML $c_t$ and the image-based GUI screenshot $v_t$. We adopt two complementary training schemes: supervised fine-tuning (SFT) and reinforcement learning (RL). For SFT, we follow the training paradigm of (Lu et al., 2025a), where each action is paired with a corresponding rationale automatically generated by Claude-3.5-Sonnet. For RL, we build on the hierarchical reward design in Shop-R1 (Zhang et al., 2025), assigning structured rewards for action prediction and self-confidence score for rationale generation, thereby enhancing the model's reasoning capabilities. Our study postprocess the raw data from the OPeRA dataset (Wang et al., 2025c), a publicly available dataset of online shopping sessions with aligned screenshots, HTML states, and action traces. To adapt OPeRA for VLM-based behavior simulation, we reorganize and preprocess the data into a task-ready benchmark. Our key **contributions** are:

- **Task-specific GUI-aware dataset construction.** We reorganize and preprocess the raw OPeRA dataset to create a benchmark tailored for simulating human online shopping behavior with VLM agents. Each input instance consists of the current webpage screenshot, the full action history, and past pruned HTML observations (retaining only elements visible in the screenshot) within the same session.

- **GUI-aware simulation of online shopper behavior.** We present, to our knowledge, the first systematic integration of textual context and visual perception for online shopper behavior simulation. Leveraging VLMs, we align agent decision-making with realistic human online shopping patterns. Experimental results show that incorporating image input alongside text improves exact match accuracy by over 6% compared to text-only baselines.

- **Revisiting limitations and envisioning futures.** We identify and discuss critical limitations in existing simulation pipelines, including action-prediction formatting, multi-modal context

fusion, long-context compression, and personalization of behavior simulation, and outline promising future research directions for each.

## 2 RELATED WORK

**LLM for human behavior simulation.** Large Language Models (LLMs) have recently demonstrated remarkable capabilities in modeling human behavior across a variety of domains. From social science simulations (Park et al., 2023a; 2024) to recommender systems (Wang et al., 2023b), and user experience (UX) research (Lu et al., 2025b), LLM-driven agents are being used to predict user actions by conditioning on interaction histories and persona attributes. These models utilize contextual cues such as user preferences, demographics, and session-based activity traces to generate contextually appropriate and personalized behavior predictions. In parallel, there has been growing interest in enhancing these simulations with explicit reasoning chains. Techniques like ReAct (Yao et al., 2023) and reflexion-based prompting (Shinn et al., 2023; Park et al., 2023b) encourage LLMs to articulate intermediate thoughts before producing actions, thus improving both interpretability and the alignment of agent decisions with human reasoning patterns. Systems including WebAgent (Gur et al., 2023) and UX-Agent (Lu et al., 2025b) advance this paradigm by structuring complex tasks into subgoals, relying on dedicated reasoning modules for better planning and control, particularly in interactive web environments. Moreover, agent-based LLM frameworks are increasingly being explored for simulating collaborative and multi-agent scenarios. Frameworks such as CoCo (Ma et al., 2024), MobileAgents (Wang et al., 2025b), and Operator (OpenAI, 2025) model complex environments where agents assume modular roles (e.g., planner, executor) and engage in cooperative reasoning (Qian et al., 2024; Luo et al., 2024). These architectures offer valuable insights into emergent behaviors and social dynamics in interactive settings. Despite recent advancements, the VLMs for simulating realistic human behaviors in web-based shopping scenarios remains largely underexplored. Existing approaches predominantly focus on text-only inputs (Lu et al., 2025a; Zhang et al., 2025), overlooking the critical role that visual context (e.g., webpage layouts, product imagery, and interface affordances) plays in shaping human decisions during online interactions. VLMs, with their ability to jointly process textual and visual modalities, offer a promising pathway to bridge this gap. By grounding agent actions in real-time visual observations of web environments, VLMs have the potential to produce behaviors that more faithfully mirror human attention patterns, preferences, and task-driven strategies. This work aims to take a step toward realizing this vision by investigating how visual grounding through VLMs can enhance the fidelity and realism of human behavior simulation in online shopping contexts.

**VLMs.** Recent advancements in Vision-Language Models (VLMs) have unlocked new capabilities across diverse multimodal tasks, including visual question answering (Liu et al., 2023b;a), visual dialogue (Wang et al., 2023a), image editing (Wang et al., 2024), and tool-augmented reasoning (Sun et al., 2024; Zheng et al., 2024). Most existing work focuses on *task completion*, where the VLM interprets visual inputs to directly solve goal-oriented problems, such as navigating web pages, generating image-based responses, or executing commands. These approaches commonly optimize for correctness or utility of outcomes, using single-turn or sequential inputs derived from the environment. In contrast, our work explores a complementary perspective: **rather than using VLMs purely for task solving, we leverage them to enrich the cognitive fidelity of** *simulated user behavior*. Specifically, we aim to align behavior generation with the visual context observed by users, modeling how visual stimuli shape human decision-making in real-world web environments. This focus is especially relevant in domains like online shopping, where user interactions are often driven by visual layouts, item appearances, and interface structure, which not fully captured by textual context alone. While prior multi-modal agents (Gupta & Kembhavi, 2023; Yang et al., 2023; Liu et al., 2024; Hong et al., 2024) have shown strong performance through either LLM- or VLM-driven control, they typically operate with explicit tool usage and target efficiency or accuracy in task execution. In contrast, our method uses visual inputs not to execute actions more effectively, but to generate more realistic human action sequences. This leads to a behavior simulator that better mimics how real users explore and interact with web interfaces, offering broader utility in applications such as user experience evaluation, digital twin modeling, and behavior forecasting. Our approach bridges the gap between vision-conditioned decision-making and personalized behavior simulation, demonstrating the potential of VLMs beyond their traditional role as perception modules for task agents.

## 3 METHODOLOGY

In this section, we first formalize the problem of simulating human behavior in web-based shopping environments. We then describe the dataset construction process tailored for Vision-Language Model (VLM) agents, followed by the training schemes designed to adapt the model for this task.

**Problem formulation.** A web shopping session can be represented as a sequential interaction trajectory consisting of multi-step user actions, denoted as $a_{1...t...N}$. At each time step $t$, the agent observes contextual information that defines the current state of the web environment. This context is captured through a simplified HTML representation, as proposed in (Lu et al., 2025c; Wang et al., 2025c; Zhang et al., 2025), which retains essential layout and content elements while filtering out irrelevant structures such as scripts and styling metadata. Complementing the HTML context, we incorporate a visual observation $v_t$ such as a screenshot of the current webpage to provide GUI-level perception. The objective of human behavior simulation is to learn a function $f$ that predicts the user's next-step rationale and action, conditioned on the cumulative interaction history and the current visual context:

$$f(c_{1...t}, a_{1...t-1}, v_t) = r_t, a_t, \tag{1}$$

where $c_{1...t}$ denotes the contextual HTML states up to step $t$, $a_{1...t-1}$ represents the sequence of past user actions, and $v_t$ provides the visual snapshot of the current webpage. The model is trained to output the next rationale $r_t$, reflecting the user's intent or reasoning, and the corresponding action $a_t$. For ease of downstream parsing and evaluation, the model output is required to be in JSON format, represented as a dictionary with two keys, *'rationale'* and *'action'*, whose values correspond to $r_t$ and $a_t$, respectively.

**Dataset construction.** We postprocess the raw OPeRA dataset (Wang et al., 2025c) to align with the requirements of VLM-based behavior simulation. Specially, the raw data in the OPeRA dataset were collected using the ShoppingFlow plugin, which records real human shopping behavior over a four-week period. In total, the dataset comprises 692 sessions from 51 unique users, yielding 28,904 real-world ⟨action, observation⟩ pairs. To ensure the task is well-defined and that sufficient information is available for model prediction, the action space is distilled into three primary categories: *'input'*, *'click'*, and *'scroll'*. Notably, sequences of consecutive *'scroll'* actions are merged into a single unified action, as the dataset does not capture visual state changes during scrolling. This limitation prevents the agent from discerning directional scroll intents (e.g., *'scroll up'* vs. *'scroll down'*). Therefore, the rationale behind scroll actions is abstracted to reflect the user's general information-seeking behavior within the visible portion of the webpage. More details about action spaces can be found in **Sec. A**. To ensure coherence between the text-based context (HTML) and the visual-based observation (screenshots), we further prune the HTML structure by retaining only elements that are present within the current visual viewport. This pruning step reduces noise, minimizes unnecessary context length, and provides a consistent alignment between textual and visual modalities. Additionally, as the original dataset contains a limited number of user-written rationales, we augment the dataset by generating rationale annotations for each action step. Specifically, we utilize Claude-3.5-Sonnet via Amazon Bedrock to synthesize plausible rationale sentences $r_t$ that capture the user's underlying motivations for performing action $a_t$. This augmentation ensures that every interaction step is paired with an interpretable reasoning trace, which is critical for training rationale-aware VLM agents.

**Training schemes.** To adapt VLMs to the task of human behavior simulation in web shopping environments, we adopt two training paradigms proposed by recent state-of-the-art LLM-based methods (Lu et al., 2025a; Zhang et al., 2025). The first approach follows the supervised fine-tuning (SFT) paradigm introduced in (Lu et al., 2025a). Here, the behavior simulation model $f$ is trained to jointly generate rationales and corresponding actions by maximizing the likelihood of annotated rationale-action trajectories. Given an input query $q_t$, which includes the contextual HTML up to step $t$ ($c_{1...t}$), past actions ($a_{1...t-1}$), past rationales ($r_{1...t-1}$), and current screen observation $v_t$, the objective is formulated as:

$$L_{\text{sft}} = -\sum_{t=1}^{N} \log p(r_t, a_t \mid q_t), \tag{2}$$

where the model learns to align its predictions with the human-annotated rationale-action pairs. This supervised learning phase establishes a strong foundation for behavior simulation by teaching the model explicit reasoning and decision-making patterns.

The second training scheme proposed by Shop-R1 (Zhang et al., 2025) utilizes reinforcement learning (RL) with hierarchical reward design and difficulty-aware reward scaling (DARS) to refine the policy. In particular, DARS scales rewards across different action types according to their relative difficulty, thereby discouraging reward hacking and encouraging more robust policy optimization. Unlike SFT, which passively mimics annotated data, RL optimizes agent behavior through tailored reward signals that promote interpretability, structured output, and task alignment. Specifically, rationale generation and action prediction are decoupled, each receiving customized rewards. First of all, to ensure model outputs remain machine-parsable and structurally valid, a binary reward signal $R_{\text{format}}$ is utilized to incentivize responses formatted in a strict JSON schema. This addresses parsing ambiguities often observed in open-ended LLM outputs. For rationale generation, a *self-certainty score* (Kang et al., 2025; Zhao et al., 2025) is computed to measure the model's confidence in its generated rationale. This score is calculated by measuring the KL divergence between the model's token-level predictive distribution and a uniform distribution:

$$s(r_t \mid q_t) = \frac{1}{N|V|} \sum_{j=1}^{N} \sum_{i=1}^{|V|} p_{ij} \log\left(\frac{p_{ij}}{U_i}\right), \qquad (3)$$

where N is the length of the generated rationale $r_t$, $p_{ij}$ denotes the predicted probability of token $i$ at position j, and $U_i = \frac{1}{|V|}$ represents a uniform distribution over vocabulary V. Higher scores correspond to more confident and coherent reasoning traces. For action prediction, the reward landscape for action prediction is shaped hierarchically. At a coarse level, correctly identifying the high-level action type (e.g., *'click'*, *'input'*, *'scroll'*) yields a base reward $R_{\text{type}}$, ensuring dense and stable policy gradients. However, additional rewards $R_{\text{subaction}}$ are unlocked only when fine-grained subaction components (e.g., clickable element or input text) are accurately predicted. This hierarchical structure discourages trivial action spamming (e.g., repeatedly issuing *'scroll'* actions) and shifts the optimization towards executing complete, meaningful action sequences. Recognizing that complex actions involving long-text or fine-grained selections are inherently harder (e.g., identifying specific product variants or form fields among thousands of candidates), the predefined value of DARS is utilized to amplify rewards for correctly predicting these challenging sub-actions. This reward scaling mechanism adjusts the reward magnitude based on task difficulty, encouraging the model to invest effort into harder but more impactful actions. Bringing these components together, the overall reward signal for reinforcement learning is formulated as:

$$R_{\text{total}} = R_{\text{format}} + s(r_t \mid q_t) + R_{\text{type}} + \text{DARS} \times R_{\text{subaction}}, \qquad (4)$$

## 4 EXPERIMENTS

**Datasets and Models.** Our experiments are conducted on the raw OPeRA dataset, which comprises 692 web shopping sessions collected from 51 unique users. Each session records multi-turn interactions between a human shopper and a website interface, capturing a sequence of user actions alongside contextual webpage states. The distribution of action types across sessions is summarized in **Tab. 1**. For contextual inputs, we

Table 1: Action type distribution within the reorganized OPeRA for the task of web shopper behavior simulation using VLMs.

| Dataset Split | *'input'* | *'click'* | *'scroll'* |
|---|---|---|---|
| Train | 499 | 4379 | 3334 |
| Test | 107 | 856 | 545 |

utilize the simplified HTML representation proposed by (Lu et al., 2025c), which preserves essential structural elements (e.g., DOM hierarchy, text nodes) while discarding irrelevant components such as scripts, styling attributes, and user-identifiable data. To ensure coherence between the textual HTML context and the corresponding visual web observations, we further prune the HTML by retaining only those elements visible within the screenshot viewport. This alignment step reduces modality mismatch and provides the model with a unified cross-modal observation space. For SFT, we augment the dataset by annotating each recorded action with a natural language *rationale*. These rationales are synthesized using Claude-3.5-Sonnet, following a carefully crafted prompting strategy detailed in Sec. B. During training, the model is tasked with producing assistant responses that contain both the rationale and a structured action prediction, conditioned on the provided interaction history (action traces and past HTMLs) as well as the current screenshot. All experiments are conducted using

Table 2: Performance comparison of next action prediction with exact match accuracy, and action type with F1 across various models, input modalities, and training configurations for the task of web shopper behavior simulation.

| Model | Input Format | Settings | Next Action Pred. Acc. | Action Type Acc. | Action Type F1 |
|---|---|---|---|---|---|
| Qwen2.5-VL-3B-Instruct | Text + Image | Zero-shot Prompt | 2.81% | 16.03% | 22.92% |
| | | SFT | 24.16% | 60.59% | 55.30% |
| | | **SFT + RL** | **44.57%** | 57.86% | 57.53% |
| | Text-only | Zero-shot Prompt | 6.41% | 34.45% | 38.79% |
| | | SFT | 20.23% | 60.86% | 53.95% |
| | | SFT + RL | 38.44% | 57.27% | 57.69% |
| | Image-only | Zero-shot Prompt | 10.81% | 44.79% | 43.82% |
| | | SFT | 19.92% | 59.31% | 53.60% |
| | | SFT + RL | 24.71% | 60.23% | 57.83% |
| Claude-3.5-Sonnet | Text + Image | Zero-shot Prompt | 9.46% | 58.64% | 45.32% |
| | Text-only | Zero-shot Prompt | 7.66% | 58.83% | 45.61% |
| | Image-only | Zero-shot Prompt | 7.95% | 60.00% | 47.04% |

the publicly available `Qwen-2.5-VL-3B-Instruct` model as the backbone. We select the 3B parameter variant, enabling practical experimentation while maintaining sufficient model capacity for multi-modal reasoning.

**Baselines for Comparison.** We compare our proposed approach against the following baseline methods: (a) **Zero-Shot Prompting**: The model is prompted to generate outputs based solely on task instructions, without any additional fine-tuning; (b) **SFT** (Lu et al., 2025a): The model is trained via supervised learning on annotated trajectories, where each action is paired with an LLM-generated rationale; (c) **SFT + RL** (Zhang et al., 2025): a RL framework that incorporates hybrid reward design to further refine simulation-oriented behavior modeling.

**Training Setups.** Our training pipelines are built upon the Qwen2.5-VL fine-tuning framework (Bai et al., 2025) for SFT, and the VERL framework (Sheng et al., 2024) for reinforcement learning. All experiments are conducted on NVIDIA A100 GPUs (80GB), utilizing Fully Sharded Data Parallelism (FSDP) in PyTorch (Zhao et al., 2023) to ensure efficient memory and compute utilization. For policy optimization, we adopt Group Relative Policy Optimization (GRPO) (Shao et al., 2024) as our default RL algorithm. Input sequences are padded or truncated to a maximum context length of 25k tokens. We employ a sampling temperature of 0.6 for generation tasks. Training is performed with a per-device batch size of 1, aggregating to a global batch size of 64 across distributed GPUs. Training hyperparameters are configured as follows: (a) for SFT: 10 epochs, learning rate of $2 \times 10^{-7}$; (b) for RL: 100 policy update steps, learning rate of $2 \times 10^{-8}$. DARS Factor is set to 10,000 by default, scaling rewards based on task difficulty.

**Evaluation Metrics.** We adopt an exact match criterion to assess the accuracy of predicted user actions. A prediction is considered correct only if all relevant components align perfectly with the ground truth. For example, in a '*click*' action, both the click subtype (e.g., filter, search bar, product option) and the target element must match. Similarly, for *input*' actions, the model must reproduce text input with equivalent semantic meaning. In addition to exact match accuracy, we report coarse-grained *action type* accuracy and F1 scores. These metrics evaluate whether the model correctly identifies the high-level action category (e.g., '*click*', '*input*', '*scroll*') regardless of fine-grained details. The comparison between exact match scores and action type metrics allows us to quantify whether residual errors arise from misclassifying the primary action type or from inaccuracies in finer-grained attributes (such as button names or input content).

**Performance analysis.** As shown in **Tab. 2**, we present a comprehensive comparison of exact match accuracy, action type accuracy, and action type F1 scores across various models, input modalities, and training regimes. Several key observations emerge from these results. First, incorporating both textual and visual inputs consistently enhances performance for the `Qwen2.5-VL-3B-Instruct` model. While zero-shot prompting with combined text and image inputs does not yield the best results, fine-tuning significantly unlocks the benefits of multi-modal grounding. This underscores the importance of aligning model representations with human decision-making processes in visually complex environments. The alignment that cannot be achieved through zero-shot prompting alone, but requires task-specific adaptation. Notably, although additional visual cues do not provide significant gains for coarse-grained action type prediction, they yield clear improvements for fine-grained

Table 3: Distribution of predicted action types (*'input'*, *'click'*, *'scroll'*, *'others'*) and invalid outputs (*'incorrect format'*) across different models, input modalities, and training settings.

| Model | Input Format | Settings | Input | Click | Scroll | Others | Incorrect Format |
|---|---|---|---|---|---|---|---|
| Qwen2.5-VL-3B-Instruct | Text + Image | Zero-shot Prompt | 2.58% | 25.72% | 6.88% | 0.08% | 64.74% |
| | | SFT | 0% | 84.36% | 15.48% | 0% | 0.16% |
| | | SFT + RL | 0% | 58.09% | 41.04% | 0% | 0.07% |
| | Text-only | Zero-shot Prompt | 1.25% | 51.95% | 12.89% | 0.39% | 33.52% |
| | | SFT | 0% | 88.20% | 11.41% | 0% | 0.39% |
| | | SFT + RL | 0% | 44.77% | 55.00% | 0% | 0.23% |
| | Image-only | Zero-shot Prompt | 5.10% | 68.80% | 25.87% | 0% | 0.23% |
| | | SFT | 0% | 89.27% | 5.87% | 0% | 4.86% |
| | | SFT + RL | 0% | 76.45% | 21.00% | 0% | 2.55% |
| Claude-3.5-Sonnet | Text + Image | Zero-shot Prompt | 2.77% | 96.56% | 0.07% | 0% | 0.60% |
| | Text-only | Zero-shot Prompt | 3.20% | 96.41% | 0.32% | 0% | 0.07% |
| | Image-only | Zero-shot Prompt | 1.32% | 97.22% | 1.39% | 0% | 0.07% |

subaction prediction, such as identifying detailed button names or input content, which rely on the model's ability to perform precise grounding and reasoning.

SFT provides substantial performance improvements across all input formats, effectively narrowing the performance gap between Text-only and Image-only modalities. After SFT, action type F1 scores rise to 53.95% for Text-only and 53.60% for Image-only inputs, indicating that both modalities, when fine-tuned on aligned action traces, can independently capture task-relevant semantics. Beyond SFT, RL further boosts model performance, particularly in exact match accuracy, which measures sequence-level consistency. For instance, the Text+Image input format achieves an exact match accuracy of 44.57% under SFT+RL, a significant jump from 24.16% under SFT alone. Similarly, Image-only exact match accuracy improves from 13.06% to 24.71%, demonstrating that RL fine-tuning enhances the model's decision precision and reduces its dependency on textual cues. Across all modalities, RL consistently pushes action type F1 scores above 57%, suggesting that its primary contribution lies in refining sequence-level alignment without compromising semantic understanding. When compared with `Claude-3.5-Sonnet`, we observe that its performance across different input modalities appears similar, exhibiting extremely low exact match accuracy but disproportionately high action type accuracy. This discrepancy arises from a strong prediction bias toward the *'click'* action, with the model often defaulting to predict *'click'* regardless of context. These results suggest that even strong closed-source models like Claude, while capable of producing outputs in the correct format as specified in the system prompt, may still underutilize cross-modal signals in structured interaction tasks unless explicitly adapted through task-aware fine-tuning. Overall, these findings highlight three critical insights: (a) multi-modal grounding is essential for aligning model predictions with human behavior in visually rich web environments; (b) SFT distills modality-specific reasoning, enabling both textual and visual inputs to capture task semantics effectively; (c) RL fine-tuning enhances sequence-level precision, ensuring coherent and high-fidelity simulation of human interaction behaviors.

**Prediction distribution analysis.** To further investigate the behavioral patterns of different models, we analyze the distribution of predicted action types, as shown in **Tab. 3**. Specifically, we categorize predictions into four main groups: *'input'*, *'click'*, *'scroll'*, and *'others'*. The *'others'* category captures outputs that fall outside the predefined action space, including ambiguous or semantically invalid actions. Additionally, we report the proportion of predictions that fail to adhere to the required structured output format, labeled as *incorrect format*. A few trends are immediately apparent. First, without any task-specific fine-tuning, all models demonstrate a substantial failure rate in producing outputs that conform to the expected structured format. This issue is especially pronounced in zero-shot settings, where the lack of explicit guidance leads to a surge in malformed or unparsable outputs. For instance, `Qwen2.5-VL-3B-Instruct` generates incorrect outputs 64.74% of the time under the Text + Image zero-shot setting, while the rate drops dramatically to under 0.2% after SFT or SFT + RL. This highlights the importance of task-specific alignment for structured output formatting. Second, action type bias differs significantly across modalities and training stages. Notably, Qwen2.5-VL-3B-Instruct exhibits a strong preference for *'click'* actions after SFT, with over 84% (Text + Image) and 88% (Text-only) of predictions falling into this category. However, with RL fine-tuning, the model adjusts toward a more realistic distribution by increasing the proportion of *'scroll'* actions, reaching 41.04% and 55.00% in the Text + Image and Text-only settings respectively. This shift suggests that RL helps calibrate action distribution to better match user interaction patterns.

Interestingly, while Image-only inputs also produce reasonably balanced action types after RL (76.45% *'click'*, 21.00% *'scroll'*), they suffer from a slightly higher formatting error rate (2.55%), indicating a potential need for further grounding visual inputs in structured generation tasks. In contrast, `Claude-3.5-Sonnet` maintains extremely low error rates even in zero-shot settings and exhibits a dominant bias toward *'click'* actions across all modalities (over 96%), but rarely predicts *'scroll'* or *'input'* actions. This further confirms that while generalist models can produce well-formed outputs, their behavioral realism is limited without task-specific training. These findings reinforce the necessity of combining supervised fine-tuning with reinforcement learning to both correct structural errors and recover realistic action distributions.

## 5 LIMITATIONS AND FUTURE DIRECTIONS

**Simulation prediction format.** Current web shopper behavior simulation tasks predominantly frame action prediction as a structured JSON generation problem, requiring models to output exact element names and action types in a parse-friendly format (Zhang et al., 2025). However, this design introduces a disconnect between human cognitive processes and model outputs. Humans rarely refer to interface elements by their DOM descriptors; instead, they rely on visual cues such as spatial location, shape, and saliency (Dardouri et al., 2024). VLMs with their capability to process visual observations offer a promising pathway to bridge this gap by enabling models to predict not only fine-grained element names but also coarse-grained spatial regions of interest within a webpage screenshot. Future datasets that record user eye-tracking data (Papoutsaki et al., 2016) or approximate attention maps during web interactions could enable more human-like simulation of attention and decision-making patterns. Such gaze-aware datasets would allow models to predict user focus areas, leading to richer simulation outputs that align more closely with real human behavior. This capability could open up new application scenarios, such as the evaluation of personalized recommender systems through offline simulations, reducing reliance on costly and slow A/B testing cycles (Rahdari et al., 2024; Shang et al., 2025).

**Multi-modal context fusion.** Existing approaches often adopt naïve concatenation strategies for multi-modal fusion, treating textual and visual contexts as independent modalities to be sequentially processed (Alayrac et al., 2022). However, images carry sparse yet spatially rich information that requires task-specific processing pipelines to extract meaningful signals. Web screenshots, in particular, are cluttered with non-informative regions such as whitespace, banners, or decorative elements, which dilute the effectiveness of simple image embeddings. Future research can consider to explore structured pipelines that include: (1) visual region detection and segmentation (Li et al., 2020), (2) semantic classification of interface components (e.g., buttons, text fields, product images), and (3) modular encoding strategies where segmented visual patches are contextually grounded and re-integrated into the HTML DOM tree. This hybrid representation can bridge textual and spatial semantics, providing models with a richer, interaction-centric context. An ambitious but plausible future direction would be to eliminate the reliance on HTML altogether, allowing VLMs to simulate web shopping behavior solely based on visual observations, akin to how humans perceive interfaces.

**Context compression.** The necessity of encoding long action histories and complex web contexts imposes significant memory and compute overhead during model training and inference. While prior works have attempted to simplify HTML structures by pruning irrelevant nodes (Lu et al., 2025c), this strategy faces an inevitable bottleneck due to the intrinsic complexity of web interfaces. A promising direction is the development of context summarization techniques that compress historical interaction sequences and user preferences into concise token sequences or latent embeddings, without sacrificing behavioral fidelity. Techniques like hierarchical memory architectures (Sun & Zeng, 2025), learned summarizers (Petrov et al., 2025), or retrieval-augmented models (Izacard et al., 2021) could be adapted to condense past context dynamically, reducing token length while retaining necessary critical decision-making cues. This is crucial for scaling behavior simulation models to real-world deployment scenarios where long-context processing remains a bottleneck.

**Personalized human behavior simulation.** One significant limitation of current datasets is the lack of longitudinal, user-specific shopping sessions. Most existing corpora aggregate behaviors across many users, modeling general human behavior rather than capturing individual idiosyncrasies (Wang et al., 2025b). Consequently, current simulations fail to reflect user-specific preferences, browsing habits, or behavioral evolution over time. Constructing large-scale, longitudinal datasets that capture the shopping trajectories of individual users over extended periods (e.g., months or years) would

enable personalized human behavior modeling. Such datasets would facilitate research in continual learning (Parisi et al., 2019), preference drift adaptation, and long-term user-agent co-adaptation. Moreover, this would allow simulation frameworks to move beyond "one-size-fits-all" models and towards agents capable of learning alongside unique users, much like personalized assistants.

## 6 APPLICATIONS

The development of realistic online shopper behavior simulators unlocks a broad spectrum of impactful applications spanning e-commerce, human-computer interaction (HCI), recommender system evaluation, and intelligent agent training. First, in customer behavior simulation for UX testing, such simulators can serve as scalable and adaptive tools for automated user experience evaluation. By capturing both the diversity and realism of human interaction patterns unlike traditional scripted bots (Wiberg & Stolterman Bergqvist, 2023) or generic LLM agents (Park et al., 2024), they enable robust stress testing of new website features, layout designs, and checkout flows under varied behavioral scenarios. Second, in personalized recommender system evaluation, synthetic but high-fidelity interaction traces can act as reliable proxies for measuring how different user personas engage with recommendation algorithms (Rahdari et al., 2024). This facilitates benchmarking of personalization quality across heterogeneous contexts while reducing dependence on costly and time-consuming A/B testing. Third, training of digital shopping assistants can directly benefit from simulators that incorporate both reasoning and action generation stages. By grounding agent decisions in multi-modal cues such as HTML context and visual observations, these assistants can be pretrained or fine-tuned to exhibit more intuitive, adaptive, and human-aligned shopping behaviors (Gur et al., 2023). Fourth, vision-language evaluation of product pages becomes feasible by integrating VLMs (Alayrac et al., 2022; Bai et al., 2025) into simulation pipelines. This allows automated assessment of how effectively product detail pages convey key attributes (e.g., discounts, usability, and product variants) through visual and textual cues, providing actionable insights for optimizing visual merchandising and page design. In summary, advances in online shopper behavior simulation promise to improve personalization, increase design efficiency, and enable the development of adaptive, user-centric systems across diverse digital commerce services.

## 7 CONCLUSION

In this work, we explored the critical role of visual perception in simulating human web shopper behavior by integrating VLMs into existing text-based simulation frameworks. Through systematic dataset construction, tailored fine-tuning strategies, and RL with structured reward design, we demonstrated that VLMs significantly enhance the fidelity of behavior simulation, particularly in visually complex e-commerce environments. Our empirical results indicate that multi-modal grounding is essential to bridge the gap between synthetic agents and real user behaviors, and that fine-tuning with task-specific supervision is crucial to fully unlock the potential of cross-modal signals. Beyond performance improvements, our study sheds light on broader methodological considerations. We highlight the importance of designing simulation paradigms that align with human cognitive processes, moving away from rigid DOM-based predictions towards visually-grounded spatial reasoning. Moreover, we advocate for more principled approaches to multi-modal context fusion, emphasizing the need for structured pipelines that can effectively disentangle and re-integrate visual and textual semantics. Addressing the challenges of context compression and personalized behavior modeling further opens avenues for future research, especially in scaling simulation frameworks to real-world applications where long-term user modeling and efficient inference are indispensable. Ultimately, this work marks a step towards more faithful and robust human behavior simulators, enabling scalable evaluation of interactive systems, such as digital assistants and recommender systems, without relying on expensive human trials. By leveraging VLMs as cognitive amplifiers, we envision a new generation of simulation frameworks that not only mimic human actions but also capture the nuanced reasoning patterns that drive real-world user interactions.

## LLM USAGE STATEMENT

Large Language Models (LLMs) were used solely as writing assistants. The authors drafted the initial versions of the paragraphs and employed an LLM to improve clarity and readability. All content was finalized through multiple rounds of human–LLM interaction, during which the authors carefully reviewed, edited, and approved the text. The research ideas, experimental design, implementation, and analysis were entirely conceived and executed by the authors without reliance on LLMs.

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

# APPENDIX

## A ACTION SPACE

```
# Action Space
An action is represented in JSON format, and there are three primary
    types of actions:

##1. 'input':
Click on an input field and type text into it.
{
    "type": "input",
    "text": "input_text"
}

## 2. 'click':
Click on a button or clickable element identified by 'name'.
It's further classified with 'click_type' including:
- 'purchase': Click on any purchase intention related buttons, including
    add cart, buy now, subscibe, checkout, etc.
- 'search': Click on search buttons or search boxes
- 'review': Click on review-related elements
- 'filter': Click on filters
- 'quantity': Click on quantity-related elements (quantity increase/
    decrease, delete item)
- 'product_option': Click on product option selections
- 'cart_side_bar': Click on shopping cart sidebar elements
- 'suggested_term': Click on suggested search terms
- 'nav_bar': Click on navigation bar elements
- 'page_related': Click on pagination elements or carousel navigation
    buttons
- 'cart_page_select': Click on cart page selection elements (e.g. item
    checkbox)
- 'product_link': Click on product links or product images
- 'other': Other types of clicks not covered by the above categories
{
    "type": "click",
    "click_type": "click_type",
    "name": "element_name"
}

## 3. 'scroll':
Scroll the page up or down for more products.
{
    "type": "scroll"
}
```

## B REASONING SYNTHESIZE PROMPT

```
<IMPORTANT>
You are given a customer's shopping journey on amazon.com. For each step,
    you will be provided with the context (what the user sees) and the
    action (what the user does). Your task is to predict the rationale
    behind the action from a first-person perspective.

Here is an example:
{example}

Output a one-sentence rationale in first person for the given action.
</IMPORTANT>
```

