# OpenReview forum: "See, Think, Act: Online Shopper Behavior Simulation with VLM Agents"
_ICLR.cc/2026/Conference — Submitted to ICLR 2026_

### Official Review · Reviewer_mQr5 · 2025-10-28

**Soundness:** 2
**Presentation:** 1
**Contribution:** 2
**Rating:** 2
**Confidence:** 4

**Summary:**

The paper addresses the problem of simulating online shopper behavior with multimodal agents. Building on the OPeRA dataset, the authors extend the traditional text-only setting by incorporating webpage screenshots, aligning them with the visible DOM elements. The proposed approach involves two stages: (1) supervised fine-tuning with rationale-augmented demonstrations, and (2) reinforcement learning with a hierarchical reward that accounts for action formatting, action-type correctness, sub-action granularity, and difficulty-aware scaling.

Experimental results show that adding visual information improves exact-match accuracy compared to text-only models, and reinforcement learning further boosts performance. Analysis highlights that vision alone already provides useful cues, though combined text+image inputs yield the best results. The paper positions its contributions as introducing multimodal shopper behavior simulation, a tailored reward design for RL, and empirical evidence of the benefits of vision-language models in this setting.

**Strengths:**

The paper addresses a practical and relevant problem of simulating online shopper behavior. It highlights the limitation of text-only modeling and makes a clear case for incorporating visual context through webpage screenshots, which is indeed closer to real user perception. The idea of aligning visual elements with the DOM structure is intuitive and helps bridge the gap between HTML-based and vision-based understanding.

The proposed training pipeline, combining supervised fine-tuning with rationale generation and reinforcement learning with hierarchical/difficulty-aware rewards, provides a reasonably structured approach. The experimental results, though limited, clearly show that adding vision improves exact-match accuracy and that reinforcement learning further enhances model performance. The analysis of action-type versus sub-action errors is also useful for diagnosing model weaknesses.

Overall, the paper contributes an initial exploration of multimodal agents for online shopping scenarios, providing a baseline that may stimulate further research in this area.

**Weaknesses:**

While the paper tackles a meaningful problem, several limitations reduce its impact:

(1) The technical novelty is limited: the approach largely combines supervised fine-tuning with rationale augmentation and a straightforward reinforcement learning reward design, without introducing fundamentally new algorithms or modeling techniques.

(2) The experimental evaluation is narrow, relying on a single dataset (OPeRA) without evidence of cross-domain generalization, ablation studies on fusion strategies, or efficiency analyses.

(3) Critical details of the reinforcement learning setup are missing, which undermines reproducibility. While the authors state that code will be released upon acceptance, this does not substitute for sufficient methodological transparency during the review process. These details should at least be provided in the appendix to ensure that reviewers and future readers can verify the validity of the approach.

(4) The task definition and positioning are somewhat unclear. While the paper frames the problem as simulating online shopper behavior, it is not well-situated within the broader landscape of web agents, recommender systems, or user modeling. This lack of clarity in both task maturity and writing presentation makes it difficult to assess the general significance of the contribution.

**Questions:**

(1) How does your proposed approach compare to existing web-agent frameworks that also use multimodal inputs (e.g., visual web navigation, GUI grounding)? More explicit positioning relative to prior work would help clarify the novelty.

(2) Have you explored cross-domain generalization beyond the OPeRA dataset? For instance, would the model trained here transfer to other e-commerce sites or more general browsing environments?

(3) Could you provide ablations on the multimodal fusion strategy (e.g., text-only, vision-only, joint) and rationale augmentation? This would make it easier to isolate the source of improvements.

(4) What is your view on the maturity of “online shopper behavior simulation” as a standalone research task? Do you see it evolving into a standardized benchmark task, or mainly as a case study of web agent capabilities?

(5) Finally, the reinforcement learning setup is not described in sufficient detail. Could you summarize the main choices (e.g., training procedure, stability measures) so that readers can better understand and reproduce the results?

---

### Official Review · Reviewer_G88H · 2025-10-29

**Soundness:** 2
**Presentation:** 2
**Contribution:** 1
**Rating:** 2
**Confidence:** 3

**Summary:**

This paper proposes a method for training a VLM-based agent that performs shopping tasks on the web. The core idea of the paper is to enable the model to generate better rationales and actions by leveraging visual information in addition to conventional HTML data. The proposed training framework consists of two stages: supervised fine-tuning (SFT) and reinforcement learning (RL). Experiments are conducted on the OPeRA dataset, comparing the proposed approach against text-only, image-only, and zero-shot prompt settings. However, no comparison with existing prior work is provided.

While the paper introduces an interesting idea of utilizing visual information for a web shopping agent, it does not present a novel approach for integrating such information. Moreover, the experiments lack comparisons with previous research, making it difficult to assess the advantage of the proposed method. The main experiments resemble ablation studies rather than comprehensive evaluations against established baselines.

**Strengths:**

- The idea of enhancing a web shopping agent’s performance by incorporating visual information is interesting and has practical potential.

**Weaknesses:**

- The justification for the reward design in Equation (4) is insufficient. Apart from the  $R_{format}$ term, the paper does not clearly explain how increasing the value of each remaining term contributes to improved model performance. It would also be helpful to clarify why the four terms are simply summed without any weighting and whether this design choice is theoretically or empirically supported.

- Although adding visual information to a VLM model for action prediction is a reasonable idea, it lacks sufficient novelty on its own.

- The experimental baselines are limited to zero-shot, SFT, and image-only configurations, without comparison to previous studies. This limitation weakens the empirical validation of the proposed approach.

**Questions:**

None

---

### Official Review · Reviewer_scDz · 2025-11-05

**Soundness:** 3
**Presentation:** 3
**Contribution:** 1
**Rating:** 2
**Confidence:** 4

**Summary:**

The key idea of the paper is that adding visual inputs (screenshots) improves behavior simulation. Using VLM trained with a standard SFT pipeline and the DARS RL algorithm (from prior work), the authors achieve the best performance on the OPeRA shopping behavior simulation benchmark.

**Strengths:**

Strengths:
- interesting, useful, and timely problem (behavior simulation)
- positive improvement on a benchmark
- well-written and clearly presented paper

**Weaknesses:**

Primary weakness: limited contribution and lack of depth

Given that the OPeRA dataset already includes screenshots, the main addition of this paper is preprocessing and filtering, which may be useful as practical implementation details but as presented, doesn’t offer new insight.

It’s quite obvious that adding visual information would help, so the scientific contribution should come from understanding how and why, not just showing a small improvement. I’m not opposed to papers that pursue an “obvious idea," but this one doesn’t dig deeper. There’s little analysis of what aspects of the visual grounding matter or how the model uses visuals. For example, here are just a few questions the authors could explore to strengthen the paper:
- what aspects of visual preprocessing matter? ablations on raw screenshot vs. preprocessing to better align with text inputs, etc.
- on different datasets (besides opera), are the visual domains different? are the design decisions different there? does training on one visual domain transfer to another?
- why does visual input help? Is it mainly because the scrolling up/down actions mentioned, or certain situations (websites with visual layouts? key steps in a user workflow?). error analysis: what does the VLM get right that the text-only systems do not?
- qualitative attention inspection: what parts of the image are important when making predictions?

The “analysis” section focuses mostly on the benefit of RL over SFT, which isn’t new and isn’t the paper’s main claim. There’s almost no discussion or evidence directly tied to the visual hypothesis.

Overall, this is a single positive result on one dataset using standard methods and an expected idea, without deeper investigation or insight, leading me to suggest a reject.

**Questions:**

See weaknesses section — would love to see some of the scientific questions addressed beyond the benchmark improvement.

---

### Official Review · Reviewer_iiSQ · 2025-11-05

**Soundness:** 1
**Presentation:** 3
**Contribution:** 1
**Rating:** 2
**Confidence:** 4

**Summary:**

This paper investigates the integration of visual information (webpage screenshots) into online shopper behavior simulation using Vision-Language Models (VLMs), building upon the OPeRA dataset. The authors propose combining textual context (HTML) and visual perception (screenshots) to simulate human shopping behavior more faithfully than text-only approaches. The framework employs supervised fine-tuning (SFT) with LLM-generated rationales and reinforcement learning (RL) with hierarchical rewards scaled by difficulty-aware factors.

**Strengths:**

1. Timely and well-motivated research direction: Incorporating visual perception into behavior simulation addresses a real gap in existing text-only approaches; GUI agent/WebNav Agent are quite trendy.

2. Reasonable experimental setup: The paper provides a clear comparison across modalities (text+image, text-only, image-only) and training schemes (zero-shot, SFT, SFT+RL).

3. Honest discussion of limitations: Section 5 provides valuable reflections on methodological constraints, including action format design, context fusion strategies, and personalization challenges.

**Weaknesses:**

In general, the novelty of this paper is very weak compared with existing VLM-based Agent work. If you look into the leaderboard of the online Mind2Web, Webarena, all of these agents adopt a very similar training pipeline. The authors didn't really understand or solve the bottleneck in this area.

1. Severely limited dataset scale and generalizability: The paper uses only 692 sessions from 51 users, resulting in just 8,212 training samples and 1,508 test samples after splitting (Table 1). This is orders of magnitude smaller than typical behavior modeling datasets and raises serious concerns about:

2. Overfitting: With such limited data, the 6% improvement could easily be noise or overfitting to the small test set
Statistical validity: No confidence intervals, significance tests, or cross-validation across different user splits
Generalizability: 51 users cannot possibly capture the diversity of real-world shopping behavior

3. Questionable contribution attribution: The paper builds heavily on existing work: SFT framework from Lu et al. (2025a), RL design from Zhang et al. (2025), GRPO algorithm from Shao et al. (2024), and data from Wang et al. (2025c). The primary novelty is "adding screenshots to existing pipelines," which is incremental. The reward structure (Equation 4) and training procedures are entirely existing ones in prior works.

**Questions:**

1. Fundamental Validity Questions

- Statistical significance: With only 51 users and 1,508 test samples, what is the variance across different random seeds or user splits? Have you performed any significance testing to confirm the 6% improvement is not due to chance?

- Overfitting evidence: Given the small dataset size, have you checked learning curves or validation performance during training? The exact match accuracy of 44.57% on such limited data raises concerns about memorization rather than learning generalizable patterns.

2. Conceptual Questions

- **Simulation fidelity** vs. **accuracy**: You optimize for exact match accuracy, but is this the right metric for simulation? A model that produces diverse, realistic-looking behaviors might be a better simulator than one that memorizes exact sequences. Have you considered diversity metrics or perplexity?

---

### Official Review · Reviewer_ywm4 · 2025-11-12

**Soundness:** 3
**Presentation:** 3
**Contribution:** 2
**Rating:** 6
**Confidence:** 4

**Summary:**

This paper models human behavior prediction in online shopping using Vision-Language Model (VLM) agents, formulating the task as a function f that predicts the action and rationale of user intent. The authors curate a <vision–text, action> dataset from the open-source OPeRA dataset, fine-tune Qwen2.5-VL using Supervised Fine-Tuning (SFT) and Reinforcement Learning (RL), and report improved performance over both text-only baselines and closed-source models (e.g., Claude-3.5-Sonnet).

**Strengths:**

(1) Thoughtful reward design for both fine and coarse-grained tasks, with structured and difficulty-aware rewards.

(2) Comprehensive evaluation metrics including exact match and F1.

(3) Strong text–vision alignment design, e.g., “We further prune the HTML structure by retaining only elements visible within the current viewport, reducing noise and aligning textual and visual modalities.” which makes a clear integration of visual context that improves grounding and simulation realism.

(4) Comprehensive section on limitations and future work

**Weaknesses:**

(1) Inconsistent abbreviation use: some acronyms are redefined after being introduced earlier.

(2) Limited real human rationales make the “human-aligned” outputs only partially aligned with actual human reasoning. How limited is the number of true human rationales?

(3) The joint prediction formulation (rationale + action) is questionable since many rationales are synthetic, potentially weakening alignment between reasoning and behavior.

(4) The statement: “Rationale generation and action prediction are decoupled, each receiving customized rewards.”
conflicts with the intuition that rationales and actions are inherently coupled, as actions are driven by rationales.

(5) Only one open-source (Qwen2.5-VL-3B) and one closed-source (Claude-3.5) model were tested — broader evaluation would strengthen claims.

(6) No ablation isolating the contribution of rationale generation (e.g., training without rationales).

(7) Human alignment and bias analysis of synthetic rationales are missing.

**Questions:**

(1) “These findings reinforce the necessity of combining supervised fine-tuning with reinforcement learning to both correct structural errors and recover realistic action distributions.”

How and why should these two be combined?
  - SFT teaches structural correctness and imitation from labeled trajectories.
  - RL optimizes high-level behavioral realism and sequence-level coherence.

However, the paper does not justify the division of labor or interaction between them.

(2) Improvements are reported, but the underlying reasons remain unclear — why did SFT help in one dimension while RL fine-tuning improved another (e.g., action distribution vs. sequence-level precision)?

(3) The authors note that:
“As the original dataset contains a limited number of user-written rationales, we augment the dataset by generating rationale annotations for each action step.”
→ Does this synthetic augmentation risk introducing bias or information leakage, especially when evaluating against human behavior (given Claude was used for rationale synthesis)?

(4) They use Claude-3.5-Sonnet via Amazon Bedrock to generate rationales for each action step.
→ Wouldn’t it be better to incorporate this rationale–action alignment directly using RLHF-style training, where a small set of real rationales supervises a human-likeness reward model?

(5) The paper claims: “Fine-tuning significantly unlocks the benefits of multimodal grounding.”
→ Is the observed improvement due to better grounding, multimodal integration, or both? Clarify this.

(6) The authors emphasize alignment with human decision-making, but they do not use RLHF or any explicit human feedback—how is “alignment” ensured in this case?

---

### Meta-Review · Area_Chair_GmR9 · 2026-01-14

**Summary:**

Multiple reviewers raised concerns about novelty, lack of comparisons, missing experiments, and narrow evaluation, with most reviewers recommending reject. There was no author rebuttal.

**Reviewer Concerns:**

The main concerns about novelty and lack of experiments, which were not rebutted.

**Reviewer Scores:**

They would have stayed the same given no rebuttal.

---

### Decision · Program_Chairs · 2026-01-26

Reject